# A Novel 0.1 mm 3D Laser Imaging Technology for Pavement Safety Measurement

**DOI:** 10.3390/s22208038

**Published:** 2022-10-21

**Authors:** Guangwei Yang, Kelvin C. P. Wang, Joshua Q. Li, Guolong Wang

**Affiliations:** School of Civil and Environmental Engineering, Oklahoma State University, Stillwater, OK 74078, USA

**Keywords:** pavement safety, pavement management system, pavement texture, pavement friction, hydroplaning speed, 3D texture parameters, artificial neural network

## Abstract

Traditionally, pavement safety performance in terms of texture, friction, and hydroplaning speed are measured separately via different devices with various limitations. This study explores the feasibility of using a novel 0.1 mm 3D Safety Sensor for pavement safety evaluation in a non-contact and continuous manner with a single hardware sensor. The 0.1 mm 3D images were collected for pavement safety measurement from 12 asphalt concrete (AC) and Portland cement concrete (PCC) field sites with various texture characteristics. The results indicate that the Safety Sensor was able to measure pavement texture data as traditional devices do with better repeatability. Moreover, pavement friction numbers can be estimated using 0.1 mm 3D data via the proposed 3D texture parameters with good accuracy using an artificial neural network, especially for asphalt pavement. Lastly, a case study of pavement hydroplaning speed prediction was performed using the Safety Sensor. The results demonstrate the potential of using ultra high-resolution 3D imaging to measure pavement safety, including texture, friction, and hydroplaning, in a non-contact, continuous, and accurate manner.

## 1. Introduction

### 1.1. Background

Road traffic crashes have caused severe social, economic, and health issues worldwide. Road traffic crash is the eighth leading cause of death globally: it kills more than 1.35 million people each year and results in another 20 to 50 million people with long-term disabilities [1]. Road traffic crashes may cost countries 2–8% of their gross domestic product every year [2]. In addition, road traffic crashes are a leading cause of death in the United States (U.S.) for people aged 1–54 and non-natural death for U.S. citizens residing or traveling abroad [3]. The U.S. suffers about 50% higher road crash deaths than similar countries in Western Europe, Canada, Australia, and Japan [2].

Therefore, safety has always been the first priority of the Federal Highway Administration (FHWA) to reduce transportation-related fatalities and serious injuries across the transportation system [4]. Crashes are always said to be multi-causal and causes are generally grouped in relation to the vehicles, road conditions, and the driver, with variable proportions [5,6]. Engineers always implement innovative technologies to improve pavement safety performance and provide safer roads to the public. Particularly, pavement texture, friction, and hydroplaning are the three main aspects when engineers performing pavement safety evaluations. Many studies have indicated that poor pavement safety conditions coincided with more traffic crashes and higher severity of traffic accidents [7,8,9,10,11,12]. Notably, pavement friction and texture have been recognized as critical surface characteristics to roadway safety in many studies: the number and severity of traffic crashes increase when roadway sections have low friction numbers or texture depth [11,13,14,15,16,17,18,19,20,21,22]. Therefore, transportation agencies have applied various devices or methods to monitor pavement friction and texture conditions as part of their asset management to reduce highway fatalities and injuries.

In addition, pavement engineers take highway design or maintenance actions to reduce the hydroplaning risk of vehicles that may lead to traffic crashes. Hydroplaning occurs when water builds up in front of a moving tire resulting in an uplift force sufficient to separate the tire from the pavement. The loss of steering and traction force produced during hydroplaning may cause the vehicle to lose control, especially when a steering tire is involved [23]. Past studies indicated the occurrence of hydroplaning is highly associated with factors including pavement texture, cross slope, longitudinal grade, pavement width, pavement types, pavement condition, tire characteristics, and rainfall intensity [24,25,26]. Consequently, highway locations with potential hydroplaning should be identified and repaired with proper treatments to minimize its possible safety risk.

### 1.2. Problem Statement

Pavement micro-texture with wavelengths less than 0.5 mm and macro-texture with wavelengths between 0.5 mm and 50 mm have been recognized as two major contributors to tire-pavement friction at different speeds [13,16]. Pavement micro-texture is typically collected through high-resolution cameras or laser devices in the laboratory environment [27,28]. High-speed data collection of micro-texture information on pavements is not possible yet as an engineering practice. Static sand patch testing or high-speed point-laser 2D texture profiler are commonly used for macro-texture evaluation via Mean Texture Depth (MTD) [29] or Mean Profile Depth (MPD) [30]. However, this 2D texture profiler only collects 2D texture profile which is challenging to distinguish texture characteristics from different pavements, as shown in Figure 1. Recently, high speed line-laser-based 3D imaging system has been applied for pavement texture data collection [31,32,33]. However, the resolution of current line-laser-based equipment is limited for accurate texture evaluation as static 3D texture testing devices. Therefore, it is desirable to collect 3D images with better resolution at a faster speed for texture evaluation.

Pavement friction data has been collected in many studies via devices such as British Pendulum Tester [34], Dynamic Friction Tester (DFT) [35], Locked-Wheel Skid Trailer [36], Grip Tester [37], or Side-Force Coefficient Routine Investigation Machine (SCRIM) [28,38,39,40,41,42,43]. However, many of these devices are expensive, hard to maintain, and unable to perform long distance friction evaluation. Further, these contact-based devices generally spray water and drag a testing tire or rubber pad across pavement surface to perform friction measurement. The variations of testing speed, water film depth, tire wear and tire pressure, and temperature will negatively affect friction results from these contact-based devices [16,44,45,46,47]. Therefore, it has been a challenge for transportation agencies to maintain an efficient and accurate management system of pavement friction using the current devices for their network level data collection.

Therefore, many studies have proposed different methods to estimate pavement friction from surface texture characteristics using non-contact based friction evaluation to eliminate the negative influence from water and rubber that are required in contact-based friction testing [48,49,50,51]. It is challenging to predict friction values based solely on MPD/MTD because the MPD/MTD was developed to evaluate simplified texture characteristics while pavement friction evaluation relies on comprehensive 3D pavement surface properties [52,53,54]. Neural network model considered 2D macro-texture profile to estimate friction numbers with good accuracy [48]. Therefore, other than traditional MTD or MPD, new texture indicators should be proposed to improve the accuracy of friction prediction models for friction evaluation via neural network model.

Lastly, empirical and analytical models have been proposed to predict pavement hydroplaning speed based on field experiments or computer simulations to identify pavement locations with hydroplaning risks. The National Aeronautics and Space Administration (NASA) hydroplaning model was proposed by Horne and Dreber to investigate tire hydroplaning based on field experimental testing [55]. Gallaway’s model was developed decades ago and is still applied to predict vehicle hydroplaning in many studies [56]. Furthermore, 2D and 3D computer simulation models were developed to evaluate pavement hydroplaning risk [26,57]. Wang et al. assessed pavement hydroplaning risk via the Inertial Measurement Unit (IMU) and texture data from 1 mm 3D imaging system [58]. However, all current studies are relevant to project level investigations, and a system that can identify pavement segments with potential hydroplaning risk from a network survey is not available yet in engineering practices.

Due to various limitations of traditional methods to evaluate pavement safety and the advancement of new high-speed 3D laser imaging technology, this study investigates the feasibility of a high-speed, non-contact, and continuous device to improve data collection and analysis for pavement safety evaluation, including pavement texture, friction, and hydroplaning speed. The newly developed 0.1 mm 3D safety sensor, referred as Safety Sensor, is the first device in the world that is possible to collect pavement 3D texture images covering macro-texture and certain range of micro-texture data at speeds up to 64 km/h (40 MPH). The successful implementation of the Safety Sensor would make network-level safety evaluation possible in a true continuous manner.

### 1.3. Objective

This study applies the Safety Sensor to evaluate pavement texture, friction, and hydroplaning in a one-shot of data collection for pavement safety evaluation. Twelve field testing sites (6 asphalt and 6 concrete sites) with different surface textures were selected for pavement texture and friction data collection. First, traditional texture and friction testing devices, including high-speed texture profiler, LS-40 pavement surface analyzer (a static 3D laser imaging device at 0.05 mm resolution), and Dynamic Friction Tester (DFT), were implemented to collect pavement texture and friction on the selected sites. Then, the Safety Sensor captured pavement 0.1 mm 2D/3D images on those sites for pavement safety evaluation. 

The MPD was calculated on 0.1 mm 3D images and compared against MPD results from LS-40 for pavement texture evaluation. Furthermore, the repeatability of the Safety Sensor and high-speed texture profiler for texture evaluation was assessed and compared. Further, eight 3D texture parameters from four classes (height, spatial, functional, and hybrid) were obtained from the 0.1 mm 3D images for friction prediction using multivariate linear regression (MLR) and artificial neural network (ANN) models. Finally, a case study of pavement hydroplaning speed prediction was performed using the Safety Sensor and IMU information.

## 2. Field Sites and Data Collection

This study selected 12 field sites to include pavement sections with different texture and friction characteristics for data collection. They included 6 asphalt concrete (AC) sites with different ages and textures, and 6 Portland cement concrete (PCC) sites with various surface finishing techniques. This study applied three traditional texture and friction testing devices on these sites for data collection. The LS-40 (Figure 2a) scanned a 114.3 mm (4.5 inches) by 101.6 mm (4 inches) area and generated 3D image with resolution of 0.01 mm for vertical direction and 0.05 mm for horizontal direction for texture evaluation in terms of MPD. The DFT (Figure 2b) measured pavement friction per the ASTM E1911 specification [35]. Further, the MPD was calculated from the high-speed texture profiler (Figure 2c) for texture evaluation per the ASTM E1845-15 specification [30] for texture repeatability evaluation. The LS-40 (Figure 2a) was used to compare the texture results from the Safety Sensor, and the DFT (Figure 2b) collected friction numbers from these sites for developing friction prediction models per 3D texture parameters from the Safety Sensor.

Lastly, the novel Safety Sensor (Figure 2d) was installed on a data collection vehicle platform to collect 0.1 mm pavement 2D/3D images with speeds up to 64 km/h (40 MPH). It uses a high-resolution camera to capture the reflection of line laser from the pavement surface to generate 2D/3D texture images based on the laser triangulation principle. More information of 3D laser imaging technology can be found in previous studies [59,60]. The Safety Sensor is 406.4 mm (16 inches) higher than the pavement surface and continuously scans a 469.9 mm (18.5 inches) wide area via 4096 pixels in a non-contact manner along left pavement wheel path. Therefore, the obtained 3D image has a resolution of 0.1 mm horizontally and 0.04 mm vertically. The 0.04 mm vertical resolution is better than the required vertical resolution of 0.05 mm for pavement macro-texture in ISO standard 13473-3. Therefore, this innovative device can be applied to mainly collect pavement macro-texture data with some micro-texture captured at highway speeds. Eight pairs of 2D and 3D images obtained from the Safety Sensor on these AC and PCC sites are illustrated in Figure 3 and Figure 4, individually. Each example image covers a 457.2 mm (1.5 ft) by 304.8 mm (1 ft) area and produces high-resolution 2D/3D pavement texture data with detailed texture characteristics.

## 3. MPD Texture Evaluation

### 3.1. MPD Compasions

The MPD is defined as the average of all the mean segment depths of all the profile segments [30]. It is a widely used indicator for pavement texture evaluation for point-laser-based equipment, such as Circular Track Meter [61] or high-speed texture profiler. Recently, the MPD has been calculated from pavement 3D images via static devices for texture evaluation [62,63]. Therefore, MPD values from the Safety Sensor will be compared against those from LS-40 on these sites to verify if the novel device is applicable for pavement texture evaluation.

The LS-40 statically collected five 3D images at 0.01 mm resolution from different spots on each site, as shown in Figure 5a. Then, the Safety Sensor continuously measured pavement 3D images at 0.1 mm resolution at a speed of 16 km/h (10 MPH) with enough length to cover the five spots measured via the LS-40. The MPDs of the five spots were calculated from 3D images of the two devices and averaged for each site to compare its similarity, as shown in Figure 5b.

The average MPD values from the Safety Sensor are comparable to those from LS-40 on most sites except Sites 1 and 4: the Safety Sensor obtained bigger MPD numbers than LS-40 on Sites 1 and 4. In Figure 5b, the trends of MPD values from these two devices are similar. The correlation coefficient of the average MPD values from these two devices is 0.98, indicating both devices can capture the texture variations for these twelve sites.

Further, for AC sites, the highest MPD (1.8 mm or 0.071 inches) was collected from Site 1, an old AC with few asphalt binders left, large aggregates exposed, and many cracks developed. The lowest MPD (0.6 mm or 0.022 inches) was collected from Site 3, a three-year-old AC parking lot with most aggregates coated by asphalt binders. The MPD values of other AC sites at different ages (Sites 2 and 4–6) are similar and range from 0.9 mm to 1.0 mm (0.035 inches to 0.040 inches). For PCC sites, the highest MPD (0.9 mm or 0.036 inches) was collected from Site 12, an old PCC with longitudinal grooves. PCC sites that are old or textured with broom or burlap drag (Sites 7–11) have an average MPD value of 0.5 mm (0.020 inches). The MPD results on these pavement sites with different textures align with observations from the collected images and expectations during field data collection. Consequently, the Safety Sensor can capture pavement texture variations at a speed of 16 km/h (10 MPH) as static texture device LS-40.

Afterward, the analysis of variance (ANOVA) test was conducted to verify whether the MPD results from the Safety Sensor is comparable to those from the LS-40. Freitas et al. used the ANOVA test to investigate pavement texture and roughness variance and repeatability [64]. Table 1 lists the ANOVA testing results of MPDs on these sites. The *p*-values of the ANOVA test on Sites 1 and 4 are <0.001 and 0.03, which are less than the significance level of 0.05. It indicates the MPD results from the Safety Sensor and LS-40 are not comparable on these two sites. The cracks on Sites 1 and 4 may cause noise in the obtained images from the Safety Sensor and generate higher MPD values than the actual results from LS-40. The *p*-values of the ANOVA test on other sites range from 0.13 to 0.95, which are larger than the significance level of 0.05. It indicates the MPD results from the Safety Sensor and LS-40 are comparable on other sites. Therefore, the Safety Sensor can perform pavement texture evaluation at speeds less or equal to 16 km/h (10 MPH) with comparable accuracy as static high-resolution texture testing devices.

### 3.2. Repeatability Evaluation

Repeatability refers to the capability of a measuring device to obtain statistically similar results from repeated runs with measuring conditions unchanged [64,65]. In this section, the repeatability of the Safety Sensor was compared against the high-speed texture profiler (Figure 2c) for texture evaluation. A 304.8 m (1000 ft) long old AC site with few cracks and another 304.8 m (1000 ft) long PCC site in good condition were selected in Oklahoma for field data collection to compare the repeatability of the two devices. Five repeated runs were performed with the high-speed texture profiler and Safety Sensor on each site at a speed of about 64 km/h (40 MPH). The MPD was calculated with an interval of 1 m (3.28 ft) for the five repeated runs.

Afterward, the correlation coefficient of MPD results for the five runs was calculated to further evaluate the repeatability of the two devices for texture evaluation. Higher correlation coefficients indicate better repeatability for that system. As listed in Table 2, both systems have correlation coefficients ranging from 0.94 to 1.00 on the PCC site, suggesting these two systems achieved comparable repeatability for texture evaluation when the pavement is in good condition. However, the correlation coefficients of MPD results are 0.50–1.00 for the high-speed texture profiler and 0.90–1.00 for the Safety Sensor on the AC site, meaning the Safety Sensor achieved better repeatability when the AC pavement is old with few cracks. Therefore, the Safety Sensor provides higher repeatability for texture evaluation than the point laser based high-speed texture profiler.

## 4. Pavement Friction Prediction

This section explores the possibility of non-contact friction evaluation with the Safety Sensor using 3D texture parameters. Pavement friction data were collected via DFT from the 12 sites. Five DFT measurements were conducted from the same spots where LS-40 measured 3D images on each site. The average friction numbers of the five DFT measurements range from 0.37 to 0.59 for AC sites, and 0.37 to 0.47 for PCC sites. These average friction numbers were used as the friction number of each site when developing friction prediction models.

On the other hand, 3D texture parameters were calculated with an interval of 1 m (3.28 ft) from the obtained 0.1 mm 3D images. In addition to MPD, another seven texture parameters, including MTD, root mean square (RMS), skewness (Ssk), kurtosis (Sku), texture aspect ratio (TAR), surface areal ratio (SAR), and surface bearing index (SBI), were calculated from the 0.1 mm 3D pavement texture images per equations in Table 3 [62]. Further, abnormal texture results from images with noise or spikes due to the vehicle’s vibration were removed before developing friction prediction models. 

Finally, 195 pairs of 3D texture parameters and DFT friction numbers (75 pairs from AC sites and 120 pairs from PCC sites) were prepared to develop friction prediction models via stepwise MLR and ANN, as listed in Appendix A. Particularly, three models were explored to investigate the relationship between pavement friction and texture features via 3D texture parameters: one model for AC sites, one model for PCC sites, and one model for both AC & PCC sites.

### 4.1. Multivariate Linear Regression Models

In this section, the 5-fold cross-validation and stepwise MLR method were used to develop the friction models. The 5-fold cross-validation shuffles and rearranges the prepared 195 datasets into five groups to minimize the bias of obtained model. The stepwise MLR was applied to develop friction models with each given resampled data by automatically selecting significant 3D texture parameters to DFT numbers and removing the texture parameters that are highly correlated or unimportant to DFT numbers during training process. Each of the five groups will be used as validation dataset in cross-validation process to minimize model’s bias and reach a general performance. 

Table 4 lists the detailed results of the three DFT models with different 3D texture parameters selected by the stepwise MLR: MTD and SAR for AC model; MPD, MTD, Skewness, and SAR for PCC model; and MTD, Skewness, TAR, and SAR for AC & PCC model. Therefore, based on the 3D texture parameters, pavement friction numbers of DFT can be predicted per Equation (9).
(9)$$\mathrm{DFT}\;\mathrm{Friction}\;\mathrm{Number}=a+\sum_{i=1}^{n}{\mathrm{Texture}}_{3D}\times b_{i}$$
where *a* is the coefficient of intercept in Table 4, ${\mathrm{Texture}}_{3D}$ is the selected 3D texture parameters in Table 4, and $b_{i}$ are the estimated coefficients for the selected 3D texture parameter ${\mathrm{Texture}}_{3D}$ in Table 4.

Further, the predicted and actual friction numbers are compared to validate the model’s performance, as shown in Figure 6a. The R^2^ values are 0.56, 0.58, and 0.34 for DFT prediction models on AC sites, PCC sites, and AC & PCC sites. These relatively low R^2^ values of friction prediction models via the stepwise MLR method indicate a nonlinear relationship between 3D texture parameters and DFT numbers.

### 4.2. Artificial Neural Network Models

Artificial neural network (ANN) models showed superior performance when addressing nonlinear regression problems including developing friction prediction models [51], even though some authors treat ANN as a “black box” because the exact relationship between the dependent variable(s) and the dependent variable(s) is unknown [43,66]. Therefore, another three friction models were developed via ANN to explore the possibilities of predicting DFT numbers via 3D texture parameters from the Safety Sensor. 70%, 15%, and 15% of the 195 samples were randomly selected for training, validation, and testing.

Figure 7 shows an example ANN structure to develop the DFT friction prediction model. The eight 3D texture parameters and one friction number were input and output of the ANN. Furthermore, 5, 10, and 15 hidden neurons were considered to identify the best network architecture for each friction prediction model. The Levenberg–Marquardt, Bayesian Regularization, and Scaled Conjugate Gradient are frequently used training algorithms in ANN structures for different applications due to their efficiency and excellent performance [51,67,68]. Therefore, the model was trained with these training algorithms to identify the best structure of ANN model.

After comparing the training results of these different combinations of training algorithms and the number of hidden neurons, the ANN model with ten hidden neurons and Bayesian Regularization obtained the best performance. The highest R^2^ values of these models are 0.95 for AC sites, 0.77 for PCC sites, and 0.84 for AC & PCC sites. The R^2^ value of the DFT model on AC sites (0.95) is higher than those on PCC sites (0.77) or AC & PCC sites (0.84). It indicates the proposed 3D texture parameters can better characterize texture properties and correlate friction numbers from DFT on asphalt pavements than concrete pavements using ANN models.

Figure 6b shows the validation results of the DFT Models via ANN models for each scenario. Comparing the validation results in Figure 6a,b, the R^2^ values of these models showed different levels of improvement with ANN implemented. For example, the R^2^ values improved from 0.56 using MLR to 0.95 using ANN on AC sites. Therefore, ANN models are more suitable to investigate the nonlinear relationship between 3D pavement texture and DFT friction numbers.

## 5. Pavement Hydroplaning Speed Prediction

This section evaluates the pavement hydroplaning speed prediction using the Safety Sensor as a case study. The Safety Sensor measures pavement texture data, and the IMU estimates pavement cross slope and longitudinal grade. The pavement hydroplaning speed is predicted via Gallaway model as following Equations (10)–(14) considering the obtained pavement texture and geometric information [69].
(10)$$S={\left(S_{l}{}^{2}+S_{c}{}^{2}\right)}^{1/2}$$
(11)$$L_{f}=W\times {\left(1+{\left({\frac{S_{l}}{S}}_{c}\right)}^{2}\right)}^{1/2}$$
(12)$$\mathrm{WFD}=0.003726\times \left[({\mathrm{MTD}}^{0.125}\times L_{f}{}^{0.519}\times I^{0.562})/\left(S^{0.364}\right)\right]-\mathrm{MTD}$$
(13)$$A=\mathrm{Max}.\;\mathrm{of}\left\{\begin{array}{c} \left(\frac{10.409}{{\mathrm{WFD}}^{0.06}}\right)+3.507\\ \left[\left(\frac{28.952}{{\mathrm{WFD}}^{0.06}}\right)-7.817\right]\times {\mathrm{MTD}}^{0.14} \end{array}\right.$$
(14)$$v_{p}:={\mathrm{SD}}^{0.04}\times P_{t}{}^{0.3}\times {\left(\mathrm{TD}+1\right)}^{0.06}\times A$$
where W: Pavement width (ft); $S_{l}$: Pavement longitudinal grade; $S_{c}$: Pavement cross slope; $L_{f}$: Pavement flow path length (ft); WFD: Water film depth (in.); MTD: Mean texture depth (in.) from the Safety Sensor; I: Rainfall intensity (in./hr); $v_{p}$: Hydroplaning speed (MPH); $P_{t}$: Tire inflation pressure (psi); SD: Spin down ratio; TD: Tire tread depth (in.).

A software tool was developed to measure pavement hydroplaning speed with user defined intervals under various circumstance, such as tire inflation pressure, rainfall intensity, and tire tread depth. The results can be exported into .csv file for other applications. 

Further, another two sites were selected as examples to illustrate the capability of the Safety Sensor in estimating pavement hydroplaning speed: Site A with HFST section (Figure 8a) and PCC section (Figure 8c) while Site B with SMA section (Figure 8b) and AC section (Figure 8d). As shown in Figure 8a–d, the HFST and SMA pavements have more texture variations than normal PCC or AC pavements. When it rains, pavement with higher texture depth provides more channels for water to escape under tire pressure to achieve desired tire-pavement traction and reduce hydroplaning risk.

The hydroplaning speeds in Figure 8 were calculated for 206.8 kPa (30 psi) tire inflation pressure, 88.9 mm/h (3.5 in./h) rainfall intensity, and 0.5 mm (0.02 inches) tire tread depth for every 1 m (3.28 ft) of pavement. A previous study indicated that the predicted driver speed is 72.4 km/h (45 MPH) when the rainfall intensity is 88.9 mm/h (3.5 in./h) [69]. Therefore, if the predicted pavement hydroplaning speed is larger than 72.4 km/h (45 MPH), there is a low hydroplaning risk for vehicles operating on the pavement. As shown in Figure 8e,f, the predicted hydroplaning speed on these two sites is larger than the expected driver speed of 45 MPH when rainfall intensity is 88.9 mm/h (3.5 in./h). It means these two sites have low hydroplaning risk. 

Particularly, on Site A, the estimated hydroplaning speeds are around 112.7 km/h (70 MPH) for PCC sections and 122.3 km/h (76 MPH) for HFST (Figure 8e). On Site B, the estimated hydroplaning speeds are around 109.4 km/h (68 MPH) for AC sections and 120.7 km/h (75 MPH) for SMA (Figure 8f). Compared with the adjacent pavements, the hydroplaning speeds are 9.6 km/h (6 MPH) higher for HFST sections on Site A and 11.3 km/h (7 MPH) higher for SMA on Site B. Therefore, the HFST or SMA sections provide higher pavement hydroplaning speeds than adjacent pavements. It indicates that the Safety Sensor is accurate when capturing pavement texture differences among treatments and can further predict corresponding hydroplaning speeds.

## 6. Conclusions and Future Studies

This study presented the initial investigation of a newly developed Safety Sensor for non-contact pavement safety measurement in terms of pavement texture, friction, and hydroplaning speed. Twelve field sites with different texture features were selected for pavement texture and friction data collection via high-speed texture profiler, LS-40, DFT, and the Safety Sensor. The MPD values from the Safety Sensor were compared against those from LS-40. The results show that the Safety Sensor was able to measure the texture variations among the 12 sites at a speed of 16 km/h (10 MPH) as the static LS-40 does. Further, the results from repeatability testing demonstrate that the Safety Sensor has higher repeatability than the 2D high-speed texture profiler for texture evaluation.

Pavement friction prediction models were also developed in the study via MLR and ANN using 3D texture parameters from four classes (height, spatial, functional, and hybrid) obtained on the 0.1 mm 3D images. In total, 195 pairs of 3D texture parameters and friction numbers were prepared for friction model development. Comparing different friction models using MLR and ANN, the following are the summarized findings:The proposed 3D texture parameters show better performance in correlating friction numbers for asphalt pavements than concrete pavements. Using ANN model as an example, the R^2^ values are 0.95 for AC sites but only 0.77 for PCC sites.ANN achieved much better accuracy in friction prediction than MLR. For example, the R^2^ values improved from 0.56 using MLR to 0.95 using ANN on AC sites.

Lastly, a case study of pavement hydroplaning speed prediction was performed using the Safety Sensor. The Safety Sensor can predict higher pavement hydroplaning speeds on HFST or SMA sections with higher texture depth than abutting regular pavement sections. The results indicate that the Safety Sensor can adequately capture pavement texture variations at high speeds, further improving the accuracy of pavement hydroplaning speeds prediction.

Therefore, it is demonstrated that the new Safety Sensor is the first device in the world that is possible to collect pavement 3D texture images covering macro-texture and certain range of micro-texture data at speeds up to 64 km/h (40 MPH). The high resolution images provide a foundation for pavement safety evaluation covering texture, friction, and hydroplaning speed prediction via one-shot of data collection in a continuous and non-contact manner. However, more efforts will be performed in the future to further validate or improve its performance:More data should be collected in the laboratory or field sites to develop MLR and ANN model.Investigate the performance of the Safety Sensor at higher testing speeds (>16 km/h (10 MPH)).Apply Deep Learning methods to estimate pavement friction based on the 0.1 mm 3D texture images when considering different testing speeds, water film depth, etc.Verify the effectiveness and consistency when using the Safety Sensor to predict pavement hydroplaning speeds in a broader study on more pavement sections.

## Figures and Tables

**Figure 1 sensors-22-08038-f001:**
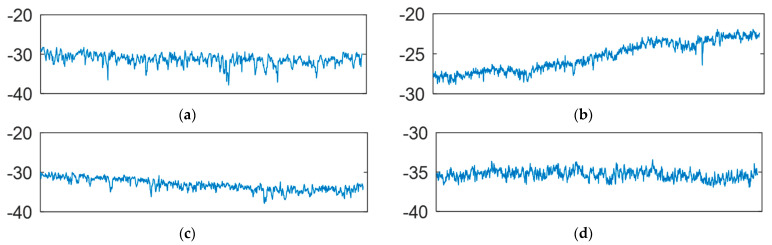
2D texture profiles of different pavements: (**a**) Stone Matrix Asphalt (SMA), (**b**) Portland cement concrete (PCC), (**c**) asphalt concrete (AC), and (**d**) High Friction Surface Treatments (HFST).

**Figure 2 sensors-22-08038-f002:**
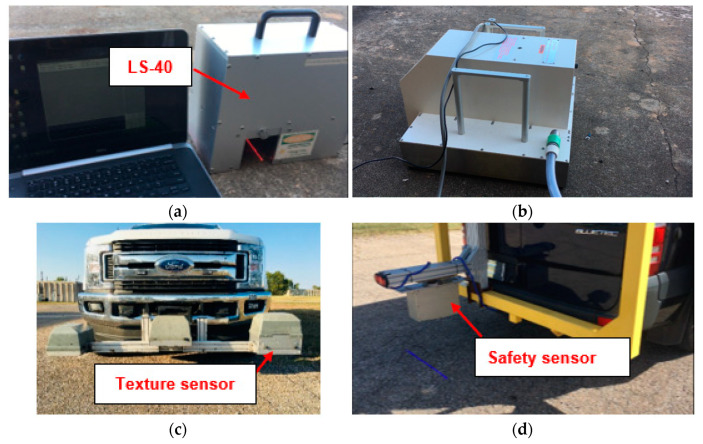
Data collection devices: (**a**) LS-40, (**b**) DFT, (**c**) High-speed texture profiler, and (**d**) Safety Sensor.

**Figure 3 sensors-22-08038-f003:**
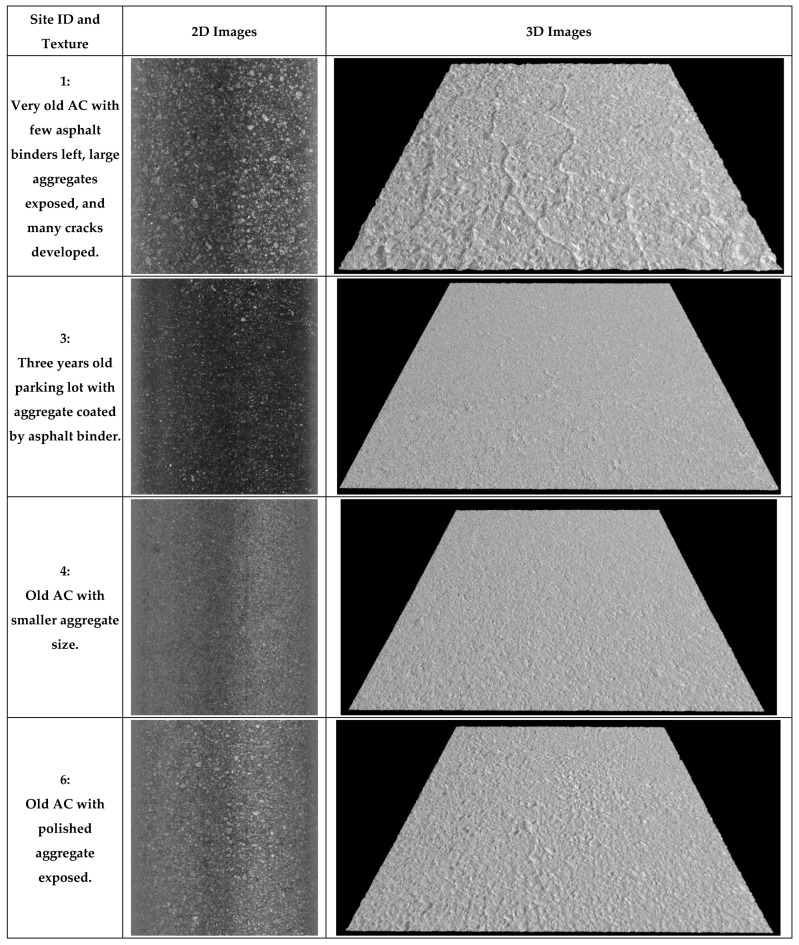
Example 0.1 mm images on AC sites.

**Figure 4 sensors-22-08038-f004:**
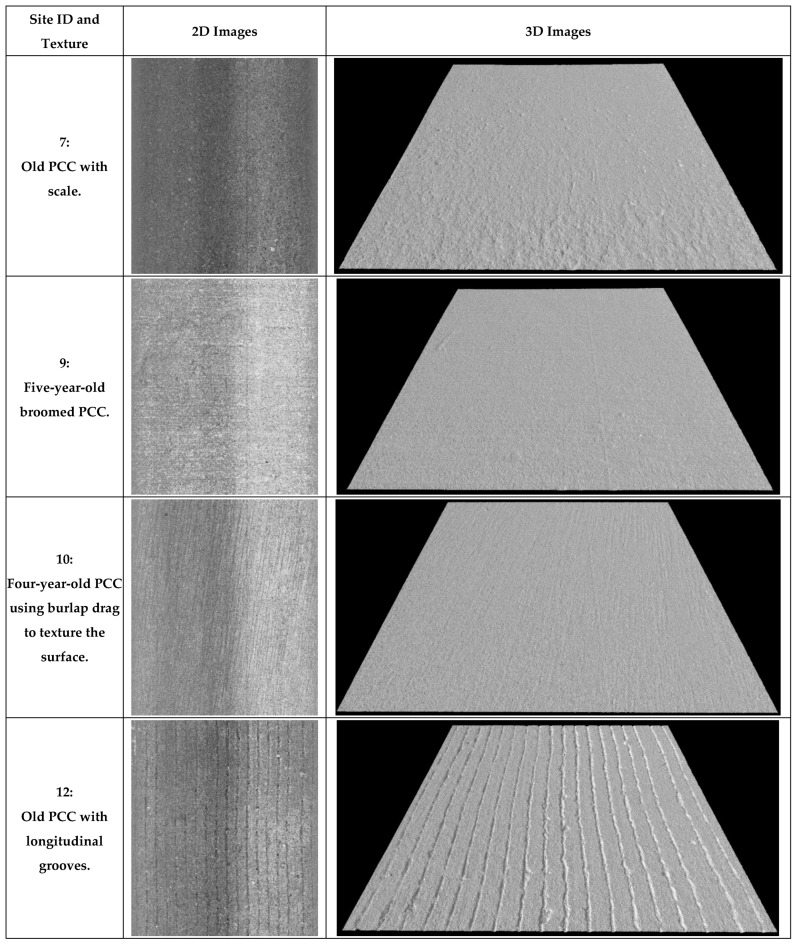
Example 0.1 mm images on PCC sites.

**Figure 5 sensors-22-08038-f005:**
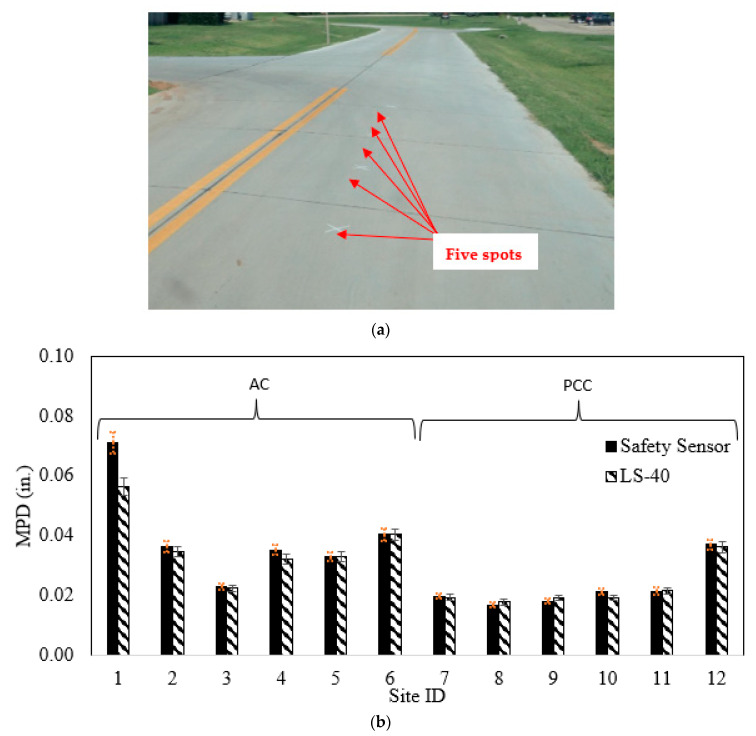
Field data collection and MPD results: (**a**) Field data collection, and (**b**) Summary of average MPD.

**Figure 6 sensors-22-08038-f006:**
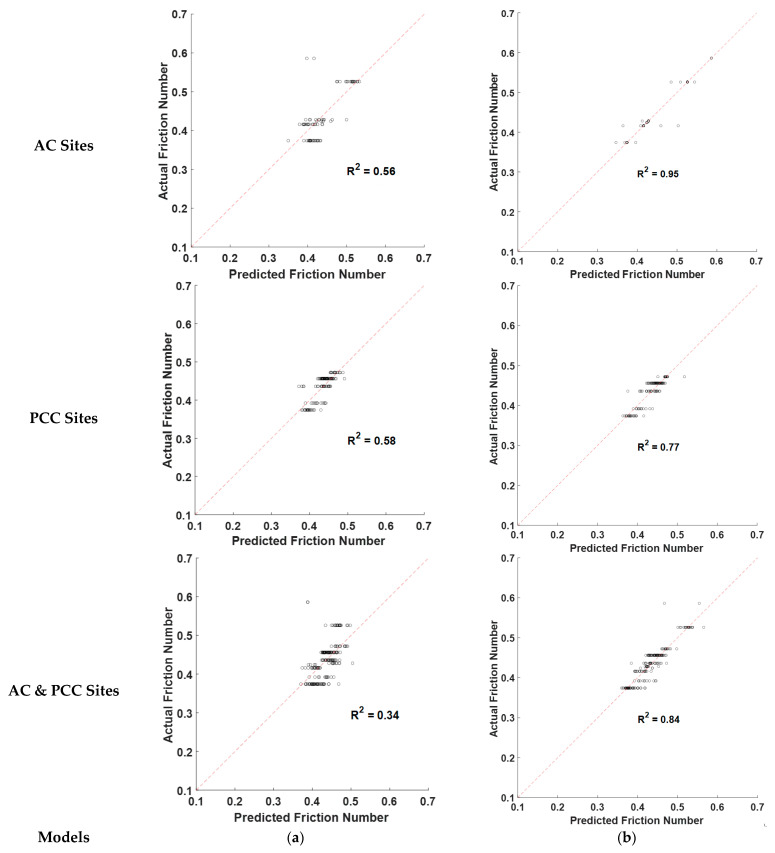
Validation results of friction prediction models: (**a**) MLR models, and (**b**) ANN models.

**Figure 7 sensors-22-08038-f007:**
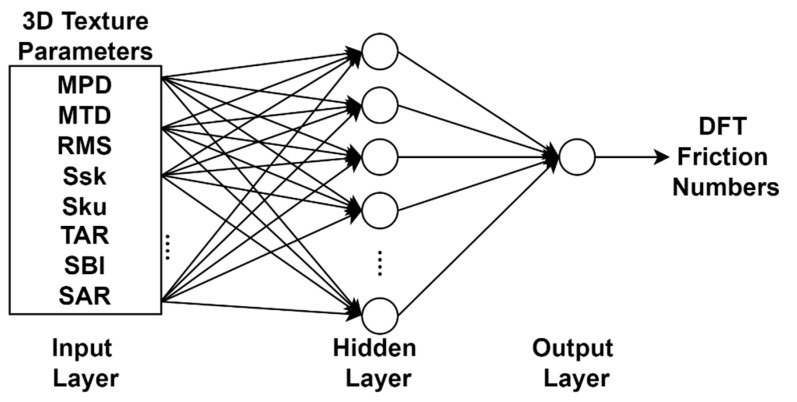
Example structure of ANN model.

**Figure 8 sensors-22-08038-f008:**
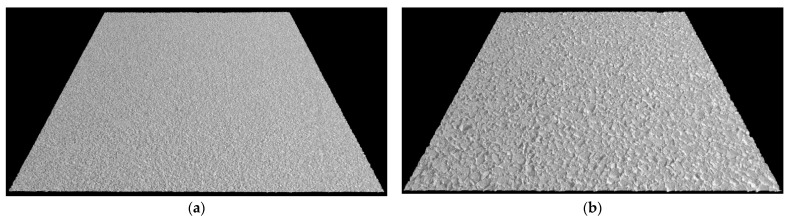
Hydroplaning speed prediction: (**a**) 0.1 mm 3D image of HFST (Site A), (**b**) 0.1 mm 3D image of SMA (Site B), (**c**) 0.1 mm 3D image of PCC (Site A), (**d**) 0.1 mm 3D image of AC (Site B), (**e**) Hydroplaning speed of Site A, and (**f**) Hydroplaning speed of Site B.

**Table 1 sensors-22-08038-t001:** ANOVA test results of MPD.

Site ID	Safety Sensor		LS-40		*p*-Value
	Average	Variance	Average	Variance	
1	0.071	2.76 × 10^−4^	0.056	9.06 × 10^−4^	<0.001
2	0.036	3.94 × 10^−5^	0.035	1.18 × 10^−4^	0.23
3	0.023	3.94 × 10^−5^	0.022	0.00 × 10	0.43
4	0.035	0.00 × 10	0.032	1.18 × 10^−4^	0.03
5	0.033	7.87 × 10^−5^	0.033	0.00 × 10	0.93
6	0.040	7.87 × 10^−5^	0.040	2.36 × 10^−4^	0.50
7	0.020	7.87 × 10^−5^	0.019	0.00 × 10	0.58
8	0.017	3.94 × 10^−5^	0.018	7.87 × 10^−5^	0.29
9	0.018	7.87 × 10^−5^	0.019	1.18 × 10^−4^	0.40
10	0.021	3.94 × 10^−5^	0.019	1.57 × 10^−4^	0.13
11	0.021	1.18 × 10^−4^	0.021	7.87 × 10^−5^	0.95
12	0.037	1.97 × 10^−4^	0.036	3.54 × 10^−4^	0.71

**Table 2 sensors-22-08038-t002:** Correlation coefficients of repeated runs.

Systems	Sites	R1-R5	R2-R5	R3-R5	R4-R5	R5-R5
High-Speed Texture Profiler	AC	0.59	0.85	0.50	0.68	1.00
	PCC	0.94	0.96	0.96	0.96	1.00
Safety Sensor	AC	0.92	0.93	0.95	0.90	1.00
	PCC	0.96	0.97	0.94	0.97	1.00

Note: the results under “R1-R5” are correlation coefficients between run No. 1 and run No. 5. A similar rule applies for “R2-R5”, “R3-R5”, “R4-R5”, and “R5-R5”.

**Table 3 sensors-22-08038-t003:** Summary of 3D texture parameters.

Categories	Parameters	Unit	Description	Equation	
Height	Mean Profile Depth (MPD)	mm	MPD is the average of all mean segment depths of all segments of the profile.	$\mathrm{MPD}=\frac{\mathrm{Peak}\;\mathrm{level}\;\left(1\mathrm{st}\right)+\mathrm{Peak}\;\mathrm{level}\;\left(2\mathrm{nd}\right)}{2}-\mathrm{Average}\;\mathrm{level}$	(1)
	Mean Texture Depth (MTD)	mm	MTD is estimated by calculating the volume and area of the 3D images.	$\mathrm{MTD}=\frac{\sum_{x=1}^{N}\sum_{y=1}^{M}\left[F_{0}-F\left(x,y\right)\right]}{D}$	(2)
	Root Mean Square (RMS)	mm	RMS is a general measurement of surface texture deviation property.	$\mathrm{RMS}=\sqrt{\frac{\sum_{x=1}^{N}\sum_{y=1}^{M}z{\left(x,y\right)}^{2}}{M\times N}}$	(3)
	Skewness (Ssk)	Unitless	Ssk represents the degree of symmetry of the surface heights about the mean plane.	$\mathrm{Ssk}=\frac{\iint_{0}^{D}\left[z{\left(x,y\right)}^{3}\right]dxdy}{S_{q}^{3}}=\frac{\sum_{x=1}^{N}\sum_{y=1}^{M}z{\left(x,y\right)}^{3}}{M\times N\times S_{q}^{3}}$	(4)
	Kurtosis (Sku)	Unitless	Kurtosis values indicate a presence of inordinately high peaks or deep valleys (Sku > 3) or lack thereof (Sku < 3).	$\mathrm{Sku}=\frac{\iint_{0}^{D}\left[z{\left(x,y\right)}^{4}\right]dxdy}{S_{q}^{4}}=\frac{\sum_{x=1}^{N}\sum_{y=1}^{M}z{\left(x,y\right)}^{4}}{M\times N\times S_{q}^{4}}$	(5)
Spatial	Texture Aspect Ratio (TAR)	Unitless	TAR is a measure of the spatial isotropy or directionality of the surface texture.	$\mathrm{TAR}=\frac{\mathrm{The}\;\mathrm{distance}\;\mathrm{that}\;\mathrm{the}\;\mathrm{normalised}\;\mathrm{ACF}\;\mathrm{has}\;\mathrm{the}\;\mathrm{fastest}\;\mathrm{decay}\;\mathrm{to}\;0.2\;\mathrm{in}\;\mathrm{any}\;\mathrm{possible}\;\mathrm{direction}}{\mathrm{The}\;\mathrm{distance}\;\mathrm{that}\;\mathrm{the}\;\mathrm{normalised}\;\mathrm{ACF}\;\mathrm{has}\;\mathrm{the}\;\mathrm{slowest}\;\mathrm{decay}\;\mathrm{to}\;0.2\;\mathrm{in}\;\mathrm{any}\;\mathrm{possible}\;\mathrm{direction}}$	(6)
Functional	Surface Bearing Index (SBI)	Unitless	SBI is the ratio of the root mean square to the surface height at a 5% bearing area.	$\mathrm{SBI}=\frac{\sqrt{\iint_{0}^{D}\left[z\left(x,y\right)\right]dxdy}}{H_{5\%}}=S_{q}$ */* $\;H_{5\%}$	(7)
Hybrid	Surface Area Ratio (SAR)	Unitless	SAR reveals the hybrid property of surfaces and is useful in applications involving surface coatings and adhesion.	$\mathrm{SAR}=\frac{\left(A-\left(M-1\right)\left(N-1\right)\times ∆x\times ∆y\right)}{\left(M-1\right)\left(N-1\right)\times ∆x\times ∆y}$	(8)

Note: the variables in Equations (1)–(8) are explained in [62].

**Table 4 sensors-22-08038-t004:** Results of DFT prediction models via MLR.

Models	Model Statistics	Model Parameters					
		Intercept	MPD	MTD	Ssk	TAR	SAR
AC Sites	Coefficient	2.13	NA	1.72	NA	NA	−24.49
	*p*-value	6.33 × 10^−19^	NA	7.84 × 10^−8^	NA	NA	3.63 × 10^−14^
	R^2^	0.56					
PCC Sites	Coefficient	2.82	2.63	1.29	−0.12	NA	−37.29
	*p*-value	4.0 × 10^−19^	9.5E-09	2.4 × 10^−2^	1.8 × 10^−5^	NA	1.6 × 10^−15^
	R^2^	0.58					
AC & PCC Sites	Coefficient	1.47	NA	1.88	−0.16	−0.23	−16.03
	*p*-value	1.7 × 10^−26^	NA	7.2× 10^−13^	9.3 × 10^−5^	3.9× 10^−3^	1.5 × 10^−15^
	R^2^	0.34					

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
