# Peer review of "A Novel 0.1 mm 3D Laser Imaging Technology for Pavement Safety Measurement"

_sensors, 2022, doi:10.3390/s22208038_

Round 1

Reviewer 1 Report

An initial investigation is presented of a recently developed sensor that can conduct non-contact pavement texture measurements. The novel device can collect 0.1 mm pavement 2D/3D images with speeds up to 64 km/h.
Generally speaking, it is an interesting device, which can help highway agencies collect pavement texture data at high speeds. Concerning the content, some points must be corrected/modified. They are commented on as they appear in the text.
Line 14. AC and PCC must be defined the first time they are used, both in the abstract and in the rest of the text.
Keywords: I suggest also adding “pavement management system” because data collection is an essential part of any Pavement Management System.
Line 37. I think that there is a gap after “system [4]”. The FHWA indeed aims to improve road safety but then, it is directly indicated that engineers are always looking for safer pavements. At this point, it should be necessary to mention the main causes of crashes. Crashes are always said to be multi-causal and causes are generally grouped in relation to the vehicles, road conditions, and the driver, with variable proportions (Studer et al. 2018, Perez-Acebo et al. 2021).
Lines 76-77. More examples from SCRIM can be included, for example, Echaveguren et al. 2010 and Echaveguren and Solminihac 2011, as it is widely used in Europe, Australia, New Zealand, and Chile. The list can be completed with the SKM (Setenkraftmessverfahren) employed in Germany, a side-force measurement device, based on SCRIM (Perez-Acebo et al. 2022).
Line 88. After indicating that many studies have proposed some methods to estimate pavement friction from surface texture characteristics, some examples are needed. Some suggestions: Yang et al. 2018; Yang et al. 2019; Yang et al. 2021, Zou et al. 2021, Zuniga-Garcia and Prozzi, 2019).
Line 113-114. In SI units, the speed in kilometer/hour is abbreviated as km/h. Please, replace all the “KPH” with “km/h”.
Figure 5 b). Include the variation range for each bar in the figure.
Table 1. The p-value cannot be 0.00. There is always a probability, even one in a million. Therefore, if the number is very low, indicate < 0.001, for example.
Line 210. After “(ANOVA) test”, indicate that the values are available in Table 1.
Line 217. Generally, “P-value” is written as “p-value”.
Table 3. The equations, even if they are presented in a table, must be numbered. Consequently, attached to the equation, a number must be included: (X) (like in line 284). Additionally, all the variables in the equations must be explained, even if they are obvious to the authors. Given that the equations are presented in a table, variables can be explained as a footnote to the table.
Line 291. Instead of “R-squared”, write R2, with the 2 as a super index.
Lines 306. At this point, after indicating that an ANN model is employed due to its superior performance, it would be convenient to say that some authors regard the ANN models as a “black box” because the exact relationship between the dependent variable(s) and independent variable(s) is unknown (Perez-Acebo et al. 2022; Gharieb et al. 2022). In fact, compared with Equation (1), we do not know the exact equation developed by the ANN model.
Lines 337-341. All these expressions must be treated (and numbered) as individual equations, not as a unique one.
Line 352. HFST and SMA must be defined.
Figure 8. It could be placed vertically. In fact, the journal’s template prefers to use vertical pages.

Echaveguren et al. (2010). Long-term behaviour model of skid resistance for asphalt roadway surfaces. Canadian Journal of Civil Engineering, 37, 719-727.

Echaveguren and de Solminihac (2011). Seasonal variability of skid resistance in paved roadways. Proceedings of the Institution of Civil Engineers – Transport, 164, TRI1, 23-32.

Gharieb et al. (2022). Modeling of pavement roughness utilizing artificial neural network approach for Laos national road network. Journal of Civil Engineering and Management, 28 (4), 261-277.

Perez-Acebo et al. (2021). Evaluation of the radar speed cameras and panels indicating the vehicles’ speed as traffic calming measures (TCM) in short length urban areas located along rural roads. Energies, 14, 8146.

Perez-Acebo et al. (2022). A simplified skid resistance predicting model for a freeway network -to be used in a pavement management system. International Journal of Pavement Engineering, DOI: 10.1080/10298436.2021.2020266

Studer et al. (2018). Analysis of the Relationship between Road Accidents and Psychophysical State of Drivers through Wearable Devices. Applied Sciences, 8, 1230.

Yang, G. et al. 2018. Convolutional neural network–based friction model using pavement texture data. Journal of Computing in Civil Engineering, 32(6), 04018052.

Yang, G. et al. 2019. Random Forest–based pavement surface friction prediction using high resolution 3D image data. Journal of Testing and Evaluation, 49(2), doi:10.1520/JTE20180937.

Yang, G. et al. 2021. Automatic pavement type recognition for image-based pavement condition survey using convolutional neural network. Journal of Computing in Civil Engineering, 35(1), 04020060.

Zou, Y., Yang, G., & Cao, M. 2021. Neural network-based prediction of sideway force coefficient for asphalt pavement using high-resolution 3D texture data. International Journal of Pavement Engineering, DOI: 10.1080/10298436.2021.1884862.

Zuniga-Garcia, N., & Prozzi, J.A. 2019. High-definition field texture measurements for predicting pavement friction. Transportation Research Record.

Author Response

Thanks very much for your time reviewing our paper. All you comments have been addressed in the revised manuscript. Please see our detailed response in the attached file "Response sensors-1956717 - reviewer 1.docx".

Reviewer 2 Report

This study presents the application of a high-resolution safety sensor to measure the pavement texture data and then utilizes the measured data to develop an ANN model for predicting the friction numbers via the 3D texture parameters. The presented system components and the adopted method are logical. However, the applicability of the proposed system should be further discussed. The details of the safety sensor should be presented to convince the reader this can achieve a resolution of 0.1mm, as stated in the manuscript. Further specific comments are listed below:

1.     Figure 1 should be improved to include the profile details and the corresponding feature pictures obtained from the pavements.

2.     LS-40 pavement surface analyzer was used as the benchmark to compare the performance of the proposed safety sensor. The resolution of this static LS-40 is 0.05mm, and the safety sensor with 0.1mm 3D image resolution. Please explain how those resolutions are derived from the device hardware setting. What are the affecting factors of this resolution, the distance between the device and the pavement, the road condition, etc.?

3.     Although the results are comparable between the two devices, there are discrepancies for sites 1 and 4. Please further discuss the difference, and comment on which one is closer to the actual results.

4.     Please provide the dataset for training the MLR and ANN. -Overall, the results shown in Figure 6 suggest that the ANN does not accurately estimate. In some cases, the estimation is not good. It can be seen that the data points in those plots are limited. I suggest the authors expand the dataset extensively and improve the data quality before training and validating the ANN model.

5.     In Figure 8, please show standard image features of Non-HFST and Non-SMA, then discuss the comparison with the features shown in (a) and (c), and explain the increase of the hydroplaning speed.

6.     The novelty and contribution of this study need to be further highlighted. As stated in the paper, the safety sensor was developed in previous studies. The obtained data is insufficient to derive a good estimation model of the frictional number. The hydroplaning speed study is preliminary, and this study's overall contribution is unclear.

Author Response

Thanks very much for reviewing our paper. Your comments are helpful for us to improve the manuscript. Please find our detailed response to your comments in the attached file "Response sensors-1956717 - reviewer 2.docx".

Round 2

Reviewer 1 Report

The authors conveniently modified most of the points commented on in the review. However, there is still one point not been adequately corrected.

It refers to point 13, according to the numbers included by authors in their responses. The authors introduced the commented references, but they did not include the idea. It is known the superior performance of Artificial Neural Network for solving nonlinear problems. Nevertheless, it must be commented that some authors regard the ANN as a “black box” because the exact relationship between the dependent variable(s) and the dependent variable(s) is unknown (Perez-Acebo et al. 2022, Gharieb et al. 2022). Additionally, I showed an example in the previous review. Authors developed Equation (1), but after developing the ANN model, there is no equation available to show! This is the reason to include this idea in the text too.

Author Response

Thanks for your time and effort in reviewing this manuscript.

Reviewer 2 Report

In the response, the authors agree that more data should be collected to develop the MLR and ANN models. I suggest that conclusions and discussions be added to address this limitation in the manuscript. The results shown now are not convincing and not applicable to field inspection. 

The dataset (195 pairs of 3D texture parameters) is strongly suggested to be included as an appendix to give credibility to this study.

Author Response

(The authors gave the same response as above.)
